# CONDENSING VIDEOS BY LEARNING WHERE MOTION MATTERS

## ABSTRACT

Video dataset condensation aims to mitigate the immense computational cost of video processing, but faces the unique challenge of preserving the complex interplay between spatial content and temporal dynamics. Prior work often unnaturally disentangles these elements, overlooking their essential interdependence. We introduce Dynamic Frame Synthesis (DFS), a novel approach that preserves this critical coupling. DFS begins with a minimal set of key frames and dynamically synthesizes new ones by identifying moments of high motion complexity, where simple interpolation fails, through gradient misalignments. This adaptive process allocates new frames only where such complexity exists, creating highly efficient and temporally coherent synthetic datasets. Extensive experiments show DFS outperforms prior methods on standard action recognition benchmarks, creating powerful representations with significantly less storage.

## 1 INTRODUCTION

Machine learning research has progressed substantially through the parallel development of novel algorithmic frameworks and the growing availability of extensive training data. In the domain of computer vision, video data represents one of the richest sources of visual information, where static content elements and temporal dynamics are fundamentally intertwined. Large-scale video datasets such as Kinetics-700 (Carreira & Zisserman, 2017), HowTo100M (Miech et al., 2019), and YouTube-8M (Abu-El-Haija et al., 2016) have enabled remarkable advances in video understanding (Carreira & Zisserman, 2017; Wang et al., 2016), from action recognition, object tracking (Bertinetto et al., 2016; Li et al., 2018), predicting future events (Farha & Gall, 2019), to realistic video generation (Tulyakov et al., 2018). However, the exponential growth of these datasets introduces substantial computational demands for storage, preprocessing, and training, creating a significant barrier that can limit broader participation in research. Just as image dataset condensation has emerged as an effective solution to this challenge in the image domain (Zhao et al., 2020; Zhao & Bilen, 2023; Zhao et al., 2023; Guo et al., 2023; Cazenavette et al., 2022), a similar need is acutely felt for video.

However, video condensation presents a fundamental challenge not found in images: the inseparable interdependence of content and motion. A pioneering work in this area by Wang et al. (Wang et al., 2024) proposed a two-stage approach that disentangles video into static content and dynamic motion. While this decomposition may offer computational advantages, it fundamentally misrepresents the nature of real-world video. For instance, in a "clapping" action, a static frame of hands already touching precludes the possibility of representing the "moving hands together" motion. Content, in this way, constrains the trajectory of motion, and motion defines the evolution of content. The key challenge, therefore, is not to model content and motion separately, but to identify the precise moments where they are most critically intertwined.

Based on this insight, we propose Dynamic Frame Synthesis (DFS), a new paradigm for condensing videos by learning where motion matters as shown in Figure 1. Instead of optimizing a fixed set of frames, DFS begins with a minimal representation, such as the start and end frames of a video. We build on the base assumption that simple motion can be adequately approximated by linear interpolation between these anchors. Our method focuses on the moments where this assumption breaks, instances of non-linear spatiotemporal change signaled by gradient misalignments. It is at these critical junctures that DFS intelligently synthesizes a new key frame, dynamically focusing its

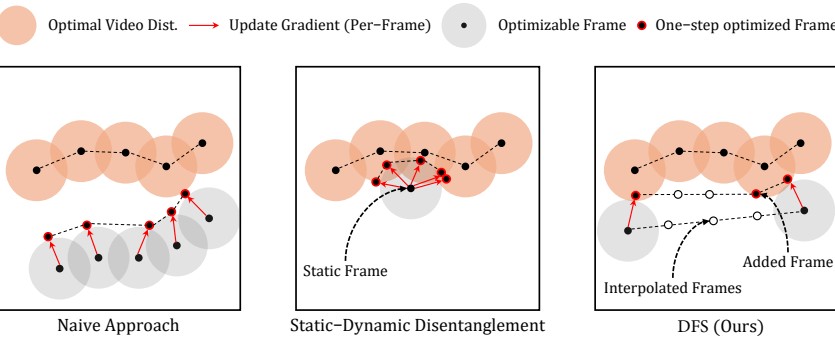

Figure 1: Visual representation of prior video dataset condensation methods and DFS (Ours). In frame-wise matching, each frame gets updated individually, neglecting the relation between frames. Static and dynamic disentangling method Wang et al. (2024) learn the temporal dynamics; however, it is restricted by the frozen pre-trained static image. Unlike these methods, our method learns the motion dynamics without any constraints to a single frame through a holistic approach.

representational capacity on the most informative parts of the action. As a result, DFS allocates more frames to complex actions while keeping simple ones compact, achieving superior performance and memory efficiency simultaneously.

Our main contributions are as follows:

- We propose DFS, a new paradigm that dynamically condenses videos by learning to identify moments of high motion complexity.
- We introduce a gradient-guided synthesis mechanism that adaptively allocates frames, preserving the crucial content-motion coupling.
- We demonstrate through extensive experiments that our method achieves state-of-the-art performance and storage efficiency across multiple action recognition benchmarks.

## 2 RELATED WORKS

### 2.1 DATASET DISTILLATION

Dataset distillation aims to synthesize a small, highly informative dataset that captures the essential characteristics of the original large-scale dataset. When models are trained on these condensed datasets, they can achieve performance comparable to training on the full dataset, but with significantly reduced computational and storage requirements. As deep learning models and datasets continue to grow in size, this field has evolved into several methodological branches.

**Gradient Matching** This approach ensures that synthetic data produces similar gradient updates as the original dataset. DC (Zhao et al., 2020) pioneered this direction by formulating dataset distillation as a bi-level optimization problem that matches single-step gradients between original and synthetic datasets. DSA (Zhao & Bilen, 2021) enhanced this framework through differentiable Siamese augmentation, improving generalization by ensuring consistent gradients across various data transformations. IDC (Kim et al., 2022) contributed efficient parameterization strategies by storing synthetic images at lower resolutions and upsampling during training, reducing storage requirements while maintaining performance. These methods provide a direct way to ensure that synthetic data induces similar training behavior as the original dataset.

**Distribution Matching** These methods aim to align feature distributions between synthetic and real data, often providing more efficient alternatives to gradient-matching. DM (Zhao & Bilen, 2023) introduced a framework that aligns distributions in embedding space, significantly improving computational efficiency. CAFE (Wang et al., 2022) ensures that statistical feature properties from synthetic and real samples remain consistent across network layers, providing more comprehensive feature alignment. Distribution-matching methods typically offer better scaling properties when condensing large-scale datasets with numerous categories.

**Trajectory Matching** Rather than matching single-step gradients or feature distributions, these methods aim to match entire training trajectories. MTT (Cazenavette et al., 2022) developed techniques to create condensed datasets by mimicking the training trajectories of models trained on the original dataset, significantly improving distillation efficiency. DATM (Guo et al., 2023) introduced difficulty-aligned trajectory matching to enable effective distillation without performance loss even as the synthetic dataset size changes. These approaches capture longer-range training dynamics, often resulting in better performance than single-step methods.

## 2.2 Video Dataset Condensation

Despite extensive research on image dataset condensation, the field of video dataset condensation remains largely unexplored, with only Wang et al. (Wang et al., 2024) making notable contributions. Their approach disentangles static content from dynamic motion by distilling videos into a single RGB static frame for content representation and a separate multi-frame single-channel component for motion. Their method follows a two-stage process: first, training the static component, then freezing it while updating only the dynamic component. Through experiments with varying numbers of condensed frames, they found that frame count does not significantly impact action recognition performance, leading to their focus on a hybrid static-dynamic representation.

Our work differs fundamentally in how it approaches the interaction between content and motion. While previous methods explicitly separate these two components by first learning a static representation and then optimizing motion as an auxiliary signal, we adopt a holistic training framework that treats the video as a fully coupled spatiotemporal structure from the beginning.

## 3 Method

Let $\mathcal{D} = \bigcup_{c=0}^{C-1} \mathcal{D}_c$ where $\mathcal{D}_c = \{(V_c^i, y_c^i)\}_{i=1}^{N_c}$, denote the real dataset consisting of $C$ classes. Each video $V_c^i \in \mathbb{R}^{T \times H \times W \times 3}$ contains $T$ frames of height $H$, width $W$ and 3 color channels. The goal of video dataset condensation is to synthesize a compact synthetic dataset:

$$S = \bigcup_{c=0}^{C-1} S_c, \quad S_c = \{(S_c^j, y_c^j)\}_{j=1}^{M_c}, \quad M_c \ll N_c, \tag{1}$$

such that each condensed video $S_c^j$ effectively captures essential spatiotemporal patterns specific to class $c$, while drastically reducing memory and computation costs with minimal degradation in downstream task performance.

### 3.1 Temporal Frame Interpolation

DFS (Dynamic Frame Synthesis) initializes each motion sequence $S_c^j$ using only the first and last frames of the video segment, rather than the entire sequence of $T$ frames. This two-frame initialization serves as a sparse temporal anchor from which new frames are progressively inserted during training. This design is motivated by prior works in video frame interpolation, which show that simple or low-velocity motion can often be approximated by linear interpolation between the two endpoints (Niklaus et al., 2017; Liu et al., 2017). We denote this initial set of frames as:

$$S_c^j = \{s_{c,1}, s_{c,T}\}. \tag{2}$$

All intermediate frames between these key frames are populated by linear temporal interpolation to construct a full sequence of length $T$. Given two adjacent key frames $s_{c,k_i}$ and $s_{c,k_{i+1}}$, the interpolated frame at time $t$ is computed as:

$$s_{c,t} = \alpha_t s_{c,k_i} + (1 - \alpha_t) s_{c,k_{i+1}}, \tag{3}$$

where $\alpha_t = \frac{k_{i+1} - t}{k_{i+1} - k_i}$ and $k_i < t < k_{i+1}$. Throughout the training, only the frames in $S_c^j$ are treated as trainable parameters.

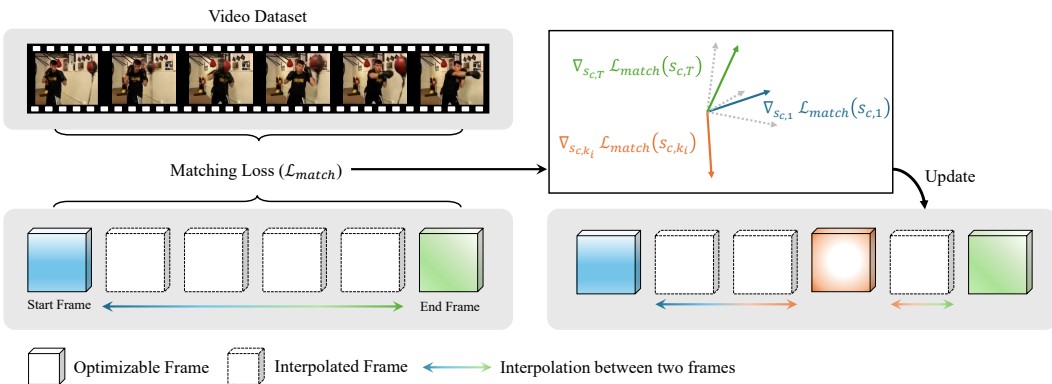

Figure 2: **Overview of DFS.** We first initialize the key frame set as the start and the end frame. As training proceeds, we calculate the cosine similarity of the gradients for each temporally interpolated frame against its two adjacent key frames. Then, the frames that have a negative correlation with its two adjacent key frames are added to the key frame set and used in the subsequent training.

## 3.2 GRADIENT-GUIDED FRAME INSERTION

Once the initial set $S_c^j = \{s_{c,1}, s_{c,T}\}$ is established, DFS progressively expands this set by inserting frames that deviate from linear motion, as indicated by their gradient directions. At each training step, we consider interpolated candidate frames $s_{c,t}$ that lie between adjacent key frames $s_{c,k_i}$ and $s_{c,k_{i+1}}$, where $k_i < t < k_{i+1}$. For each candidate frame, we compute its gradient $\nabla \mathcal{L}(s_{c,t})$ and measure its cosine similarity with the gradients of its two adjacent key frames:

$$\cos_i^t = \cos\left(\nabla\mathcal{L}(s_{c,t}), \nabla\mathcal{L}(s_{c,k_i})\right), \tag{4}$$

$$\cos_{i+1}^t = \cos\left(\nabla\mathcal{L}(s_{c,t}), \nabla\mathcal{L}(s_{c,k_{i+1}})\right). \tag{5}$$

If both cosine similarities are below $\epsilon$, i.e.,

$$\cos_i^t < \epsilon \quad \text{and} \quad \cos_{i+1}^t < \epsilon, \tag{6}$$

then the candidate frame $s_{c,t}$ is considered to be the frame that is at the position of which the motion cannot be represented through linear interpolation and is inserted into the key frame set:

$$S_c^j \leftarrow S_c^j \cup \{s_{c,t}\}. \tag{7}$$

This gradient-based criterion captures non-linear transitions in appearance or motion and enables the model to refine its support set by inserting only those frames that contribute meaningful learning signals. The insertion process is repeated iteratively throughout training except for the warm-up and cool-down phases, resulting in a temporally adaptive sequence that emphasizes semantically rich regions.

Our frame insertion strategy is theoretically justified through the following lemmas.

**Lemma 1 (Loss-Descent Blockage under Gradient Misalignment)** *Let* $s_t = \alpha s_{k_i} + (1 - \alpha)s_{k_{i+1}}$, *with* $0 < \alpha < 1$, *be a linear interpolation between two key frames* $s_{k_i}$ *and* $s_{k_{i+1}}$. *Let the task-loss gradients be*

$$g_t = \nabla_{s_t}\mathcal{L}(s_t), \ g_i = \nabla_{s_{k_i}}\mathcal{L}(s_{k_i}), \ g_{i+1} = \nabla_{s_{k_{i+1}}}\mathcal{L}(s_{k_{i+1}}).$$

*Suppose* $\langle g_t, g_i \rangle < 0$ *and* $\langle g_t, g_{i+1} \rangle < 0$. *Then, for any convex combination*

$$v = \lambda(-g_i) + (1-\lambda)(-g_{i+1}), \quad \lambda \in [0,1],$$

*it holds that*

$$\langle g_t, v \rangle > 0.$$

*Consequently, no first-order update to only the endpoints decreases* $\mathcal{L}(s_t)$. *Thus,* $s_t$ *must be promoted to the key-frame set for further loss minimization.*

*Proof.* See Appendix. □

Table 1: Experiment results on two video benchmarks and prior methods categorized into coreset methods, static/dynamic disentangled methods, and holistic methods. [†] represents the author provided results. Higher values are better. **Bold** and underline denote the best and second-best scores for each setting, respectively.

| Method | MiniUCF | | | HMDB51 | | |
|---|---|---|---|---|---|---|
| | VPC 10 | VPC 5 | VPC 1 | VPC 10 | VPC 5 | VPC 1 |
| **Coreset Methods** | | | | | | |
| Random | $27.8_{\pm 1.1}$ | $19.6_{\pm 0.4}$ | $10.9_{\pm 0.7}$ | $9.8_{\pm 0.4}$ | $6.8_{\pm 0.7}$ | $3.3_{\pm 0.1}$ |
| Herding | $\mathbf{33.7}_{\pm 0.3}$ | $26.3_{\pm 1.0}$ | $13.2_{\pm 1.3}$ | $10.8_{\pm 0.6}$ | $9.0_{\pm 0.6}$ | $3.0_{\pm 0.1}$ |
| K-Center | $29.1_{\pm 0.6}$ | $23.2_{\pm 0.7}$ | $13.9_{\pm 1.6}$ | $8.0_{\pm 0.1}$ | $5.2_{\pm 0.4}$ | $2.4_{\pm 0.4}$ |
| **Static / Dynamic Disentangled Methods** | | | | | | |
| DM | $30.0_{\pm 0.6}$ | $25.7_{\pm 0.2}$ | $15.3_{\pm 1.1}$ | $12.1_{\pm 0.4}$ | $8.0_{\pm 0.2}$ | $6.1_{\pm 0.2}$ |
| Wang et al.[†] | - | $27.2_{\pm 0.4}$ | $17.5_{\pm 0.1}$ | - | $8.2_{\pm 0.1}$ | $6.0_{\pm 0.4}$ |
| **Holistic Method** | | | | | | |
| **DFS** | $31.0_{\pm 0.1}$ | $\mathbf{28.0}_{\pm 0.1}$ | $\mathbf{17.9}_{\pm 0.3}$ | $\mathbf{12.8}_{\pm 0.2}$ | $\mathbf{10.5}_{\pm 0.4}$ | $\mathbf{7.5}_{\pm 0.3}$ |
| Whole Dataset | | $57.8_{\pm 1.1}$ | | | $25.4_{\pm 0.2}$ | |

## 3.3 WARM-UP AND COOL-DOWN PHASE

To ensure stable frame insertion dynamics during training, we introduce a warm-up and cool-down phase.

**Warm-Up Phase.** During the early stage of training, we disable gradient-guided frame insertion and optimize only the initial key frames in the set $S_c^j = \{s_{c,1}, s_{c,T}\}$. This allows the endpoints to be optimized before being used as reference anchors for gradient-guided frame insertion. Without this phase, premature insertion based on noisy gradients may lead to redundant or suboptimal key frame selection.

**Cool-Down Phase.** In the final training phase, frame insertion is again suspended. Since newly inserted frames near convergence would receive insufficient updates, they may remain underoptimized and degrade downstream performance. By freezing the key set toward the end of training, we ensure that all frames in the set receive adequate supervision.

## 3.4 OPTIMIZATION OBJECTIVE

Let $f_\theta$ denote a feature extractor network, and let $\mathcal{B}_c^{\text{real}}$ and $\mathcal{B}_c^{\text{syn}}$ be the real and synthetic video batches for class $c$, respectively. Each synthetic video $S_c^j$ contains a subset of trainable key frames, while the remaining frames are linearly interpolated.

We define our optimization objective as minimizing the feature distribution discrepancy between real and synthetic samples as following:

$$\min_{\{S_c^j\}} \sum_{c=1}^C \left\| \frac{1}{|\mathcal{B}_c^{\text{real}}|} \sum_{x \in \mathcal{B}_c^{\text{real}}} f_\theta(x) - \frac{1}{|\mathcal{B}_c^{\text{syn}}|} \sum_{s \in \mathcal{B}_c^{\text{syn}}} f_\theta(s) \right\|_2^2 \tag{8}$$

During training, gradients are backpropagated only through the active key frame set $S_c^j$. Warm-up and cool-down phases regulate when new key frames are inserted, ensuring stable optimization and sufficient updates across all selected frames. The overall framework of the DFS is illustrated in Figure 2.

## 4 EXPERIMENT

### 4.1 DATASET

We conduct experiments on 4 datasets: UCF101 (Soomro et al., 2012) and HMDB51 (Kuehne et al., 2011) for small scale datasets, and Kinetics (Carreira & Zisserman, 2017) and Something-Something

Table 2: Storage requirements on two video benchmarks and prior methods categorized into coreset methods, static/dynamic disentangled methods, and holistic methods. [†] represents the author provided results. Lower values are better. The value in parentheses in indicates the size of the synthetic dataset as a percentage of the full dataset size. **Bold** and underline denote the best and second-best scores for each setting, respectively.

| Method | MiniUCF | | | HMDB51 | | |
|---|---|---|---|---|---|---|
| | VPC 10 | VPC 5 | VPC 1 | VPC 10 | VPC 5 | VPC 1 |
| **Coreset Methods** | | | | | | |
| Random | $1150_{(11.7\%)}$ | $586_{(6.0\%)}$ | $115_{(1.2\%)}$ | $1150_{(23.3\%)}$ | $586_{(11.9\%)}$ | $115_{(2.3\%)}$ |
| Herding | $1150_{(11.7\%)}$ | $586_{(6.0\%)}$ | $115_{(1.2\%)}$ | $1150_{(23.3\%)}$ | $586_{(11.9\%)}$ | $115_{(2.3\%)}$ |
| K-Center | $1150_{(11.7\%)}$ | $586_{(6.0\%)}$ | $115_{(1.2\%)}$ | $1150_{(23.3\%)}$ | $586_{(11.9\%)}$ | $115_{(2.3\%)}$ |
| **Static / Dynamic Disentangled Methods** | | | | | | |
| DM | $1150_{(11.7\%)}$ | $586_{(6.0\%)}$ | $115_{(1.2\%)}$ | $1150_{(23.3\%)}$ | $586_{(11.9\%)}$ | $115_{(2.3\%)}$ |
| + Wang et al.[†] | - | $455_{(4.6\%)}$ | $94_{(1.0\%)}$ | - | $455_{(9.2\%)}$ | $94_{(1.9\%)}$ |
| **Holistic Methods** | | | | | | |
| **DFS** | $\mathbf{324}_{(3.3\%)}$ | $\mathbf{133}_{(1.4\%)}$ | $\mathbf{20}_{(0.2\%)}$ | $\mathbf{287}_{(5.8\%)}$ | $\mathbf{137}_{(2.8\%)}$ | $\mathbf{22}_{(0.4\%)}$ |
| | 9.81GB | | | 4.93GB | | |

Table 3: Experiment results on two large-scale video benchmarks. [†] represents the author provided results. **Bold** and underline denote the best and second-best scores for each setting, respectively.

| Method | Kinetics-400 | | SSv2 | |
|---|---|---|---|---|
| | VPC 5 | VPC 1 | VPC 5 | VPC 1 |
| **Coreset Methods** | | | | |
| Random | $5.5_{\pm 0.2}$ | $3.0_{\pm 0.2}$ | $3.6_{\pm 0.1}$ | $3.1_{\pm 0.1}$ |
| Herding | $6.3_{\pm 0.2}$ | $3.3_{\pm 0.1}$ | $3.6_{\pm 0.1}$ | $2.8_{\pm 0.1}$ |
| K-Center | $6.2_{\pm 0.2}$ | $3.1_{\pm 0.1}$ | $\mathbf{4.5}_{\pm 0.1}$ | $2.6_{\pm 0.2}$ |
| **Static / Dynamic Disentangled Methods** | | | | |
| DM | $\mathbf{9.1}_{\pm 0.9}$ | $\underline{6.3}_{\pm 0.0}$ | $\underline{4.1}_{\pm 0.0}$ | $3.6_{\pm 0.0}$ |
| + Wang et al.[†] | $7.0_{\pm 0.1}$ | $\underline{6.3}_{\pm 0.2}$ | $3.8_{\pm 0.1}$ | $\mathbf{4.0}_{\pm 0.1}$ |
| **Holistic Method** | | | | |
| **DFS** | $\underline{8.1}_{\pm 0.1}$ | $\mathbf{7.1}_{\pm 0.1}$ | $\underline{4.1}_{\pm 0.1}$ | $\underline{3.9}_{\pm 0.2}$ |
| Whole Dataset | $30.3_{\pm 0.1}$ | | $23.0_{\pm 0.3}$ | |

V2 (Goyal et al., 2017) for large scale datasets. UCF101 consists of 13,320 video clips of 101 action categories. Following the prior work (Wang et al., 2024), we leverage the miniaturized version of UCF101, hereinafter miniUCF, which includes the 50 most common action categories from the UCF101 dataset. HMDB51 consists of 6,849 video clips of 51 action categories. Kinetics-400 has videos of 400 human action classes and Something-Something V2 has 174 motion-centered classes.

For miniUCF and HMDB51, we sample 16 frames per video with a sampling interval of 4 and resize frames to $112 \times 112$. For Kinetics-400 and Something-Something V2, we sample 8 frames per video and resize to $64 \times 64$. Consistent with prior work (Wang et al., 2024), we only apply horizontal flipping with 50% probability as the sole data augmentation strategy.

## 4.2 EXPERIMENTAL SETTING

For all of the experiments, we employ miniC3D, which comprises 4 Conv3D layers, as our backbone architecture following the pioneering work in video dataset condensation. Unlike DM (Zhao & Bilen, 2023), DFS is initialized from Gaussian noise rather than initializing to a random real frame from the dataset. We report the mean of three evaluations for each experiment, measuring top-1 accuracy for miniUCF and HMDB51, and top-5 accuracy for Kinetics-400 and Something-Something V2. We compare our method against three coreset selection methods (random selection, Herding (Welling, 2009), and K-Center (Sener & Savarese, 2017)), an image dataset condensation methods (DM (Zhao & Bilen, 2023)), and a video dataset condensation method (Wang et al. (Wang et al., 2024)) the first and the only video dataset condensation method. We evaluate performance

Table 4: Cross-architecture results on MiniUCF with 1 VPC. $^\dagger$ indicates author-provided results. **Bold** denotes best scores per model.

| Method | Evaluation Model | | |
|---|---|---|---|
| | ConvNet3D | CNN+GRU | CNN+LSTM |
| DM | $15.3_{\pm 1.1}$ | $9.9_{\pm 0.7}$ | $9.2_{\pm 0.3}$ |
| Wang et al.$^\dagger$ | $17.5_{\pm 0.1}$ | $12.0_{\pm 0.7}$ | $10.3_{\pm 0.2}$ |
| DFS (Ours) | $\mathbf{17.9}_{\pm 0.3}$ | $\mathbf{18.9}_{\pm 0.8}$ | $\mathbf{18.2}_{\pm 1.3}$ |

under different condensation ratios, measured as Videos Per Class (VPC). Note that the VPC follows the notation of Images Per Class (IPC) in image dataset condensation and that DFS, in most cases, will have fewer frames than 16 frames, as we are only adding frames when required. During inference, we leverage the index position for each saved vector, which is saved with the frames, with negligible memory consumption. Our experiments employ the SGD optimizer with a momentum of 0.95 for all methods. The $\epsilon$ is set to 0. The hyperparameters, including the learning rates, are detailed in the Appendix.

### 4.3 RESULTS

Table 1 presents the experiment results categorized as coreset methods, static and dynamic disentangled methods, and the holistic method. We categorized DM (Zhao & Bilen, 2023) as the disentangled methods as they initialize the frames from the real frames in the dataset. The results showcase that DFS achieves state-of-the-art performance in most experimental settings. The performance increments of DFS are smaller in the motion-centric large dataset, where only 8 frames are used for training, as shown in Table 3. However, we note that we always achieve the second-best performance, if not first. Additionally, DFS scales along with the VPC which was not the case with Wang et. al (Wang et al., 2024).

As the storage footprint of condensed data is a critical factor in dataset condensation, we report the storage requirements in Table 2. We follow the same calculation procedure as prior work (Wang et al., 2024), treating each sample as a `float32` tensor. For DFS, the reported storage corresponds to the total number of frames retained after the condensation process completes for each VPC setting. We ignore the negligible overhead from storing frame indices.

Unlike previous methods that begin with a fixed number of frames (e.g., 16), DFS starts with only 2 key frames per video and progressively inserts additional frames only when the cosine similarity between gradients is negative. This selective strategy results in a significantly lower storage footprint while achieving superior performance. Moreover, since DFS adds frames adaptively rather than proportionally to the number of VPCs, its storage does not grow linearly with VPC. This behavior is clearly visible in the miniUCF results, where storage grows much more slowly than would be expected under proportional expansion. Such efficiency makes DFS especially advantageous when users wish to scale up performance under higher VPC budgets without incurring prohibitive storage costs.

## 5 ABLATION

**Cross-Architecture** As dataset condensation aims to perform well not only on the training model but also on other architectures, we validate our approach's robustness through Table 4. The experimental results demonstrate that DFS not only achieves state-of-the-art performance compared to prior methods but also maintains robust performance across different architectures. Notably, while DM and Wang et al. (Wang et al., 2024) show significant performance drops when evaluated on CNN+GRU, our method maintains consistent performance with only minimal degradation. This strong cross-architecture generalization underscores the strength of our holistic design, which preserves the intrinsic coupling between content and motion—an essential property of video data often overlooked by prior methods.

**Effect of Number of Initial Key Frames** Table 5 presents results when varying the number of initial key frames for the condensation process. We observe a consistent decrease in performance as

Table 5: Results by varying initial representative frames of DFS. Bold denotes best scores per dataset.

| Dataset | Number of Initial Key Frames | | | | |
|---|---|---|---|---|---|
| | 2 | 3 | 4 | 6 | 8 |
| MiniUCF | $\mathbf{17.9}_{\pm 0.3}$ | $17.3_{\pm 0.8}$ | $17.0_{\pm 0.1}$ | $15.5_{\pm 0.2}$ | $15.3_{\pm 0.2}$ |
| HMDB51 | $\mathbf{7.5}_{\pm 0.3}$ | $6.8_{\pm 0.1}$ | $6.6_{\pm 0.1}$ | $6.1_{\pm 0.3}$ | $5.6_{\pm 0.2}$ |

Table 6: Ablation study results for (A) with and without insertion, (B) frame selection strategy, (C) similarity metric, and (D) training phase (warm-up and cool-down).

| Dataset | w/ Insertion | w/o Insertion |
|---|---|---|
| HMDB51 | $7.5_{\pm 0.3}$ | $6.1_{\pm 0.3}$ |
| miniUCF | $17.9_{\pm 0.3}$ | $15.8_{\pm 1.2}$ |

(A)

| Dataset | Negative Grad. | Random Pos. |
|---|---|---|
| HMDB51 | $7.5_{\pm 0.3}$ | $6.8_{\pm 0.2}$ |
| miniUCF | $17.9_{\pm 0.3}$ | $16.8_{\pm 0.4}$ |

(B)

| Dataset | Cosine Sim. | L2 Distance |
|---|---|---|
| HMDB51 | $7.5_{\pm 0.3}$ | $6.0_{\pm 0.6}$ |
| miniUCF | $17.9_{\pm 0.3}$ | $15.7_{\pm 0.7}$ |

(C)

| Dataset | w/o Warm-Up | w/o Cool-Down |
|---|---|---|
| HMDB51 | $6.8_{\pm 1.2}$ | $6.3_{\pm 0.3}$ |
| miniUCF | $16.1_{\pm 0.8}$ | $16.9_{\pm 1.3}$ |

(D)

the number of initial frames increases. This result highlights a central principle of our approach that it is not the amount of input frames that matters, but the ability to insert new frames only when they are truly needed. Starting with a minimal number of frames (i.e., 2) and selectively adding more based on gradient signals enables the model to focus on semantically meaningful motion without being distracted by redundant or misaligned updates. The performance drop seen with more initial frames supports this idea by showing that naïvely including more content can actually interfere with the optimization process. We hypothesize that starting with too many frames creates conflicting gradient signals in the early training stages, hindering the optimization of a coherent motion trajectory. In contrast, our start-small approach allows the model to first establish a robust and simple path between two anchors before refining more complex, non-linear dynamics.

**Without Frame Insertion**   One of DFS's core contributions is the progressive insertion of frames based on gradient cues. To isolate the effect of this component, we perform an ablation where only two key frames (the first and last) are optimized throughout training. Temporal interpolation is still applied between these endpoints, but no additional key frames are inserted. As shown in Table 6(A), removing this progressive insertion leads to a substantial performance drop, confirming that our insertion strategy is critical for capturing complex motion and structure. Nevertheless, this reduced variant still outperforms several existing baselines. This suggests that even a minimal version of our method can serve as a competitive and meaningful baseline for video dataset condensation.

**Frame Selection Strategy**   We evaluate the effectiveness of our gradient-based frame selection by comparing it against a random selection baseline. Both methods operate under identical conditions: a new frame is inserted whenever a negative cosine similarity is detected between gradients. However, while DFS selects the frame with the most negative cosine similarity, the baseline instead randomly selects one from the candidate pool, including the negatively correlated one. This setup ensures that the two methods differ only in how the new frame is selected, not in how often frames are added or when. As shown in Table 6(B), replacing our targeted frame selection with random addition results in substantial performance drops across both datasets. This confirms that gradient correlation is not just a useful signal but a decisive factor in identifying semantically meaningful frames that enhance the condensation process.

**Cosine Similarity vs. L2 Distance**   DFS uses cosine similarity to identify frames whose gradients are directionally misaligned with those of existing key frames, signaling potential discontinuities in motion or content. This angle-based criterion is particularly effective for capturing semantic transitions, as it not only measures the degree of difference but also the directional disagreement

between gradients. To test whether cosine similarity is truly essential, we compare it against a distance-based alternative using the L2 norm. However, unlike cosine similarity, which has a well-defined geometric interpretation, L2 distance lacks a natural threshold. To make the comparison meaningful, we calibrated the L2 threshold to 0.141, which corresponds to 10% of the unit vector distance implied by a 90-degree angular separation in cosine space. As shown in Table 6(C), cosine similarity significantly outperforms L2 distance across both HMDB51 and miniUCF. These results confirm that directional disagreement, rather than magnitude alone, is a more reliable indicator of frame-level semantic variation, justifying our use of cosine similarity for frame insertion.

**Effect of Warm-Up and Cool-Down Phases** To stabilize training and prevent premature or noisy frame insertions, DFS incorporates both a warm-up and a cool-down phase. The warm-up phase delays the start of frame insertion to allow gradients to stabilize around the initial key frames. The cool-down phase, on the other hand, suspends further insertions once the condensation process nears convergence, preventing overfitting or unnecessary growth in the synthetic video set. To evaluate the necessity of each phase, we conduct ablations where either the warm-up or cool-down phase is removed. As shown in Table 6(D), removing either phase degrades performance, with the absence of the warm-up phase causing unstable optimization due to early gradient noise, and the absence of the cool-down phase leading to over-insertion and noisy representation. These findings confirm that both the warm-up and cool-down stages are integral to DFS's temporal curriculum, ensuring effective and stable condensation dynamics throughout training. All qualitative results can be found in the Appendix.

**Optical Flow Result** To qualitatively assess whether the frames selected through our gradient-based criterion capture meaningful temporal dynamics, we visualize the class-wise mean optical flow on the mini-UCF dataset. Figure 3 compares the average optical flow of real videos (top) with that of our condensed data generated using only negatively correlated frames (IPC = 5, bottom) for the class *Soccer Penalty*. Despite the aggressive frame reduction, our method produces motion patterns that closely resem-

miniUCF → Class : Soccer Penalty

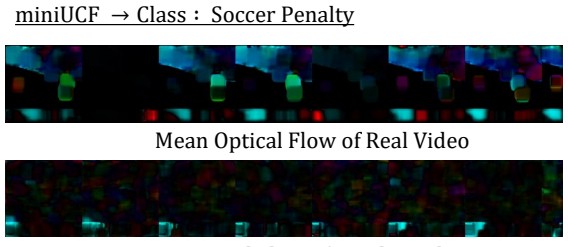

Mean Optical Flow of Real Video

Mean Optical Flow of Condensed Data

Figure 3: Optical flow comparison - Soccer Penalty class.

ble those of the real videos. This result implies that selecting frames based on gradient misalignment is not only computationally principled but also semantically grounded—our approach reliably detects the frames responsible for key motion events, even without any supervision. More qualitative results are provided in the Appendix.

## 6 CONCLUSION

In this paper, we introduced Dynamic Frame Synthesis (DFS), a novel paradigm for video dataset condensation that learns to identify and preserve the critical interplay between content and motion. Unlike prior methods that rely on a fixed representation or artificial disentanglement, DFS dynamically synthesizes keyframes where they are most needed, guided by gradient misalignments that signal high motion complexity. This adaptive strategy allows DFS to allocate representational capacity intelligently, resulting in condensed datasets that are not only compact but also temporally coherent. Our extensive experiments demonstrate that DFS sets a new state-of-the-art, outperforming existing methods in both accuracy and storage efficiency across standard action recognition benchmarks.

**Limitations** While DFS enables efficient and adaptive frame selection under typical spatiotemporal conditions, it may face challenges in two regimes. First, in videos with extremely fast or abrupt motion, linear interpolation may fail to capture dynamics. Second, for very long sequences, optimization from Gaussian noise can become unstable. Moreover, DFS is primarily designed for classification tasks, and its ability to preserve fine-grained semantic cues required for generative tasks (e.g., captioning, video-text alignment, video generation) remains unexplored.

ETHICS STATEMENT

The primary focus of this work is on improving the computational efficiency of training video models, with the goal of making research more accessible and environmentally sustainable. However, we acknowledge that any dataset condensation method carries potential ethical implications. The primary ethical consideration for DFS is the potential for inheriting and amplifying biases present in the original large-scale datasets. Since DFS aims to capture the essential data distribution, any demographic, social, or representational biases in the source data will likely be reflected or even concentrated in the condensed set. We urge practitioners using DFS to perform thorough bias audits on the source datasets and the models trained on the condensed results. Additionally, as with any technology that lowers the barrier to training powerful models, there is a risk of misuse in applications.

REPRODUCIBILITY STATEMENT

To ensure full reproducibility of our results, we include source code in the supplementary zip file. All hyperparameters required to reproduce our experiments, including learning rates, batch sizes, and the warm-up/cool-down schedules for each dataset and VPC setting, are provided in the Appendix. The data preprocessing and evaluation protocols are consistent with those of prior work to ensure fair and direct comparison.

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

# A APPENDIX

## A.1 PROOF OF LEMMA 1

**Lemma 1 (Loss-Descent Blockage under Gradient Misalignment)**

*Let $s_t = \alpha s_{k_i} + (1 - \alpha)s_{k_{i+1}}$, with $0 < \alpha < 1$, be a linearly interpolated frame between two key frames $s_{k_i}$ and $s_{k_{i+1}}$. Let the task-loss gradients be denoted as*

$$g_t = \nabla_{s_t}\mathcal{L}(s_t), \quad g_i = \nabla_{s_{k_i}}\mathcal{L}(s_{k_i}),$$

$$g_{i+1} = \nabla_{s_{k_{i+1}}}\mathcal{L}(s_{k_{i+1}}).$$

*Suppose*

$$\langle g_t, g_i \rangle < 0 \quad and \quad \langle g_t, g_{i+1} \rangle < 0.$$

*Then, for every convex combination*

$$v = \lambda(-g_i) + (1 - \lambda)(-g_{i+1}), \quad \lambda \in [0, 1],$$

*the following inequality holds:*

$$\langle g_t, v \rangle > 0.$$

*Consequently, no first-order update obtained by modifying only the two endpoint frames can decrease $\mathcal{L}$ at $s_t$; the loss is stationary or strictly increasing along every such direction. Therefore, $s_t$ must be promoted to the key frame set and directly optimized to enable further loss minimization.*

*Proof.* By the bilinearity of the inner product,

$$\langle g_t, v \rangle = \lambda\langle g_t, -g_i \rangle + (1 - \lambda)\langle g_t, -g_{i+1} \rangle.$$

Applying the assumption $\langle g_t, g_i \rangle < 0$, we obtain

$$\langle g_t, -g_i \rangle = -\langle g_t, g_i \rangle > 0,$$

and similarly,

$$\langle g_t, -g_{i+1} \rangle = -\langle g_t, g_{i+1} \rangle > 0.$$

Therefore,

$$\langle g_t, v \rangle = \lambda \cdot \langle g_t, -g_i \rangle + (1 - \lambda) \cdot \langle g_t, -g_{i+1} \rangle > 0.$$

This shows that the directional derivative of $\mathcal{L}$ at $s_t$ along any direction $v$ formed by adjusting only the endpoints is positive:

$$D_v\mathcal{L}(s_t) = \langle \nabla\mathcal{L}(s_t), v \rangle > 0.$$

Thus, no first-order update along such directions can reduce the loss at $s_t$, and $\mathcal{L}(s)$ is strictly increasing along all directions spanned by $-g_i$ and $-g_{i+1}$. It follows that further loss minimization requires directly optimizing $s_t$ as a key frame. $\square$

### A.2 HYPERPARAMETER

In Table A, we show the learning rate and batch size under each dataset and IPC. The $\epsilon$ is set to 0 for all experiments throughout the manuscript. The warm-up and cool-down phases are processed for 20% of the whole iteration each. In other words, if the condensation process is set to 100 iterations, the warm-up phase takes up the first 20 iterations and the cool-down phase takes up the last 20 iterations, leaving 80 iterations for the progressive refinement and insertion of frames. We follow the setting from the prior method (Wang et al., 2024) for evaluation and cross-architecture evaluation.

Table A: Hyperparameters for DFS under different datasets and IPC.

| Method | Dataset | Train | | | Evaluation | |
| --- | --- | --- | --- | --- | --- | --- |
| | | IPC | LR | Batch Real | Epoch | LR |
| DFS | MiniUCF | 1 | 1 | 64 | | |
| | | 5 | 25 | 64 | | |
| | | 10 | 50 | 64 | | |
| | HMDB51 | 1 | 0.7 | 64 | 500 | $1e^{-2}$ |
| | | 5 | 25 | 64 | | |
| | | 10 | 75 | 64 | | |
| | Kinetics-400 | 1 | 1 | 64 | | |
| | | 5 | 50 | 128 | | |
| | SSv2 | 1 | 3 | 64 | | |
| | | 5 | 30 | 128 | | |

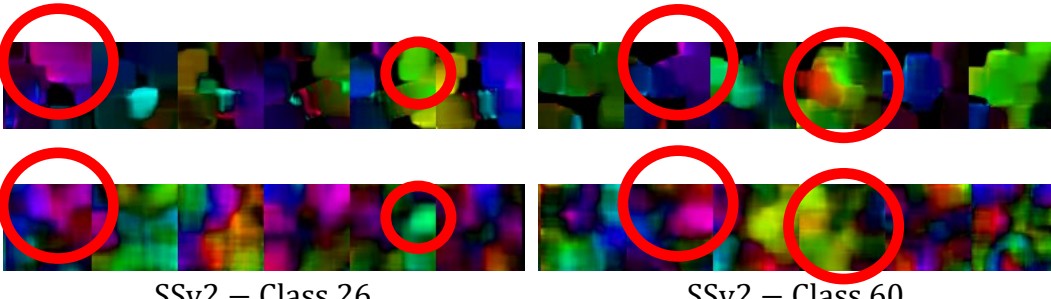

SSv2 − Class 26    SSv2 − Class 60

Figure A: Extra optical flow results on SSv2 under 1 VPC setting. The red circle shows where the optical flow matches the most between the condensed video and the real video.

### A.3 OPTICAL FLOW ANALYSIS

To further assess the temporal fidelity of our condensation framework, we visualize the optical flow fields of both real and condensed videos. Optical flow represents the pixel-wise motion between consecutive frames and serves as a direct indicator of whether the synthesized frames preserve realistic temporal dynamics. In our visualizations in the supplementary and also in the main manuscript, we use a standard HSV-based color encoding, where the hue (i.e., the color itself) corresponds to the direction of motion—such as rightward appearing reddish, leftward bluish, and upward greenish—while the saturation and brightness encode the magnitude of motion, with brighter and more saturated regions indicating stronger or faster motion. Regions with little to no motion appear desaturated or grayish.

Despite beginning from Gaussian noise and adding frames along training, the optical flow results show that DFS is capable of progressively aligning the synthesized motion with that of the real video. As shown in Figure B, the red circles highlight regions where the direction and magnitude of the condensed optical flow closely resemble those of the original video. This further supports the observation that DFS can synthesize coherent motion dynamics from sparsely supervised temporal supervision.

Nevertheless, some failure modes are also apparent in these optical flow visualizations. In cases where the first and last frames contain minimal or no motion, the model struggles and generates meaningless or abrupt motion during warm-up phase. Moreover, in action classes that involve fast or abrupt motions, the resulting flow fields from the condensed video occasionally lack directional consistency and show spatial noise, indicating poor alignment. These limitations appear to be exacerbated by the use of Gaussian noise initialization, which may hinder the model's ability to focus solely on the informative motion patterns at early training stages.

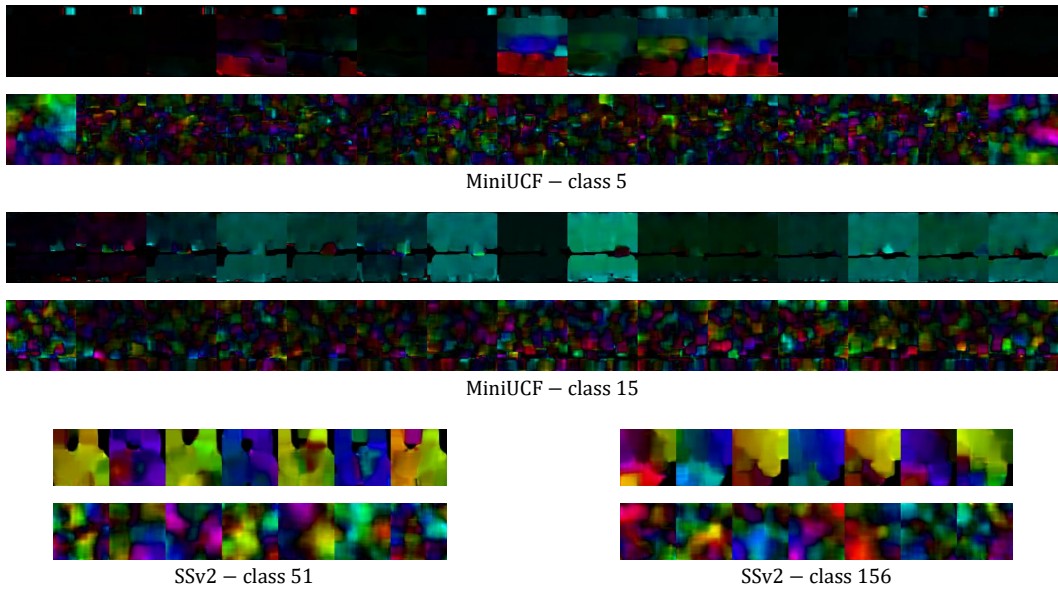

MiniUCF − class 5

MiniUCF − class 15

SSv2 − class 51                    SSv2 − class 156

Figure B: Analysis on when DFS fails.

### A.4 QUALITATIVE RESULTS

We visualize the condensed videos on HMDB51 and MiniUCF under the 1 VPC setting for maximal clarity. The visualized frames in Figure C and Figure D correspond to those retained after the condensation process, where the noise images are placeholders which does not get stored along with condensed data.

Red rectangles highlight the negative effect when the warm-up phase is omitted. As consistently observed across both datasets, removing the warm-up leads to excessive frame selection, resulting in redundant and less informative synthetic frames while consuming more memory.

Blue rectangles indicate frames produced when the cool-down phase is omitted. Although overall results appear more stable than in the warm-up-removed case, we observe that some frames are added during the final few iterations of condensation. These late-added frames often lack sufficient training, reducing their utility for action recognition by being not fully trained.

We used a Large Language Model (LLM) to assist with improving the clarity, grammar, and organization of the text. All scientific contributions, including the core methodology, experimental design, and analysis of results, are solely the work of the authors. ]The Use of Large Language Models (LLMs)

We used a Large Language Model (LLM) to assist with improving the clarity, grammar, and organization of the text. All scientific contributions, including the core methodology, experimental design, and analysis of results, are solely the work of the authors.     You may include other additional sections here.

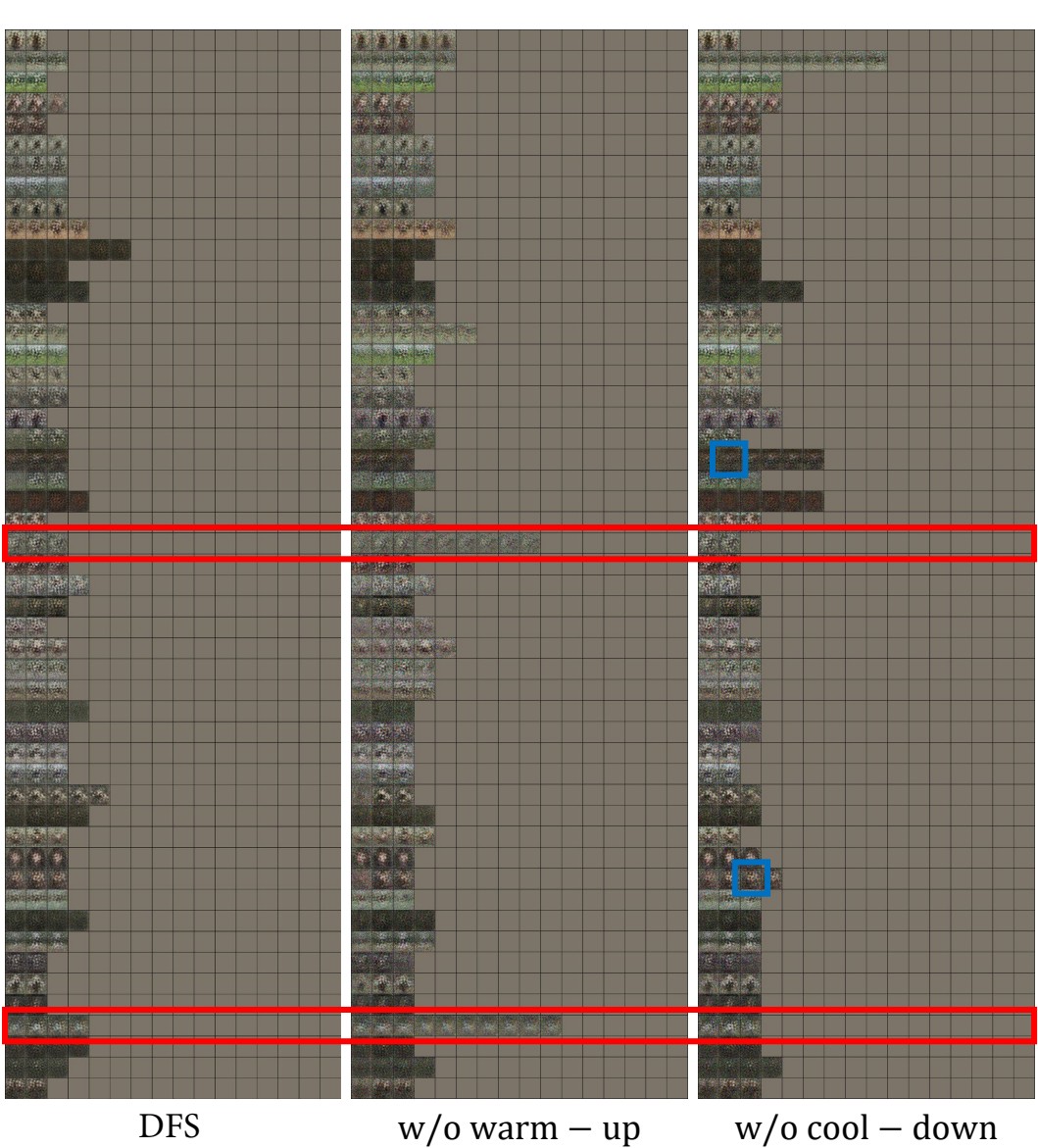

Figure C: Visualization of DFS, DFS without warm-up, and DFS without cool-down on HMDB51 under 1 VPC. Red rectangles highlight the negative effects of omitting the warm-up phase, while blue rectangles indicate frames that may be under-trained due to the absence of a cool-down phase.

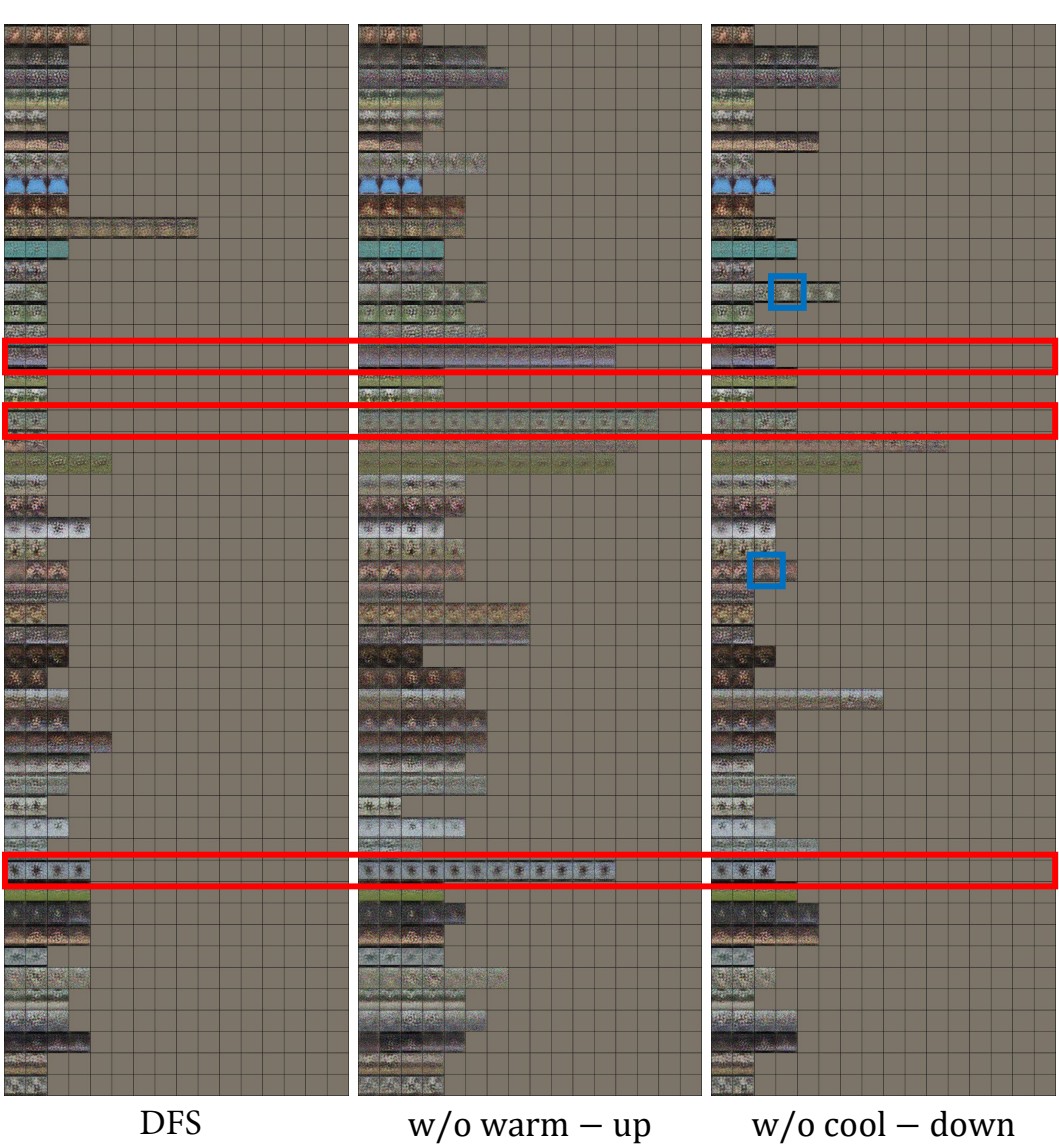

Figure D: Visualization of DFS, DFS without warm-up, and DFS without cool-down on MiniUCF under 1 VPC. Red rectangles highlight the negative effects of omitting the warm-up phase, while blue rectangles indicate frames that may be under-trained due to the absence of a cool-down phase.

