# OpenReview forum: "Condensing Videos by Learning Where Motion Matters"
_ICLR.cc/2026/Conference — ICLR 2026 Conference Withdrawn Submission_

### Official Review · Reviewer_u5pu · 2025-10-24

**Soundness:** 2
**Presentation:** 3
**Contribution:** 1
**Rating:** 4
**Confidence:** 3

**Summary:**

This paper deals with the topic of video condensation and specifically the interdependence of content and motion. They propose Dynamic Frame Synthesis (DFS) a method for video condensation that synthesizes video keyframes at times where linear interpolation breaks. They start with the first and last frame and adaptively syhtesize frame at temporal locations where the gradient

**Strengths:**

The paper deals with the interesting topic of video condensation and proposes a method that is based on a clever, intuitive idea: adding to the "summary" frames that cannot be captured via linear interpolation of the existing ones.

**Weaknesses:**

1) The experiments are only on short, curated, single action classification datasets. This is a very basic scenario for our times, and it is not clear if and how their method could be extended in a less toy scenario. This would be the equivalent of CIFAR for image classification.

2) The method assumes that the first and last frames are highly important (for every class). This makes precise trimming crucial.

3) Whether a frame can or cant be represented through interpolation in Eq (6) is based on a threshold on the cosine similarity of the gradients. This is highly heuristic and sensitive. It is unclear to me why the same epsilon should apply everywhere. The authors do not explore this.

4) It is unclear to me what Lemma 1 offers in practice. The intuition is right, but whether "further loss minimization" is important is highly arbitrary.

5) Comparisons are to very basic baselines and to only one other related work, Wang et al, a very simple method that decouples motion and appearence, representing the latter with a single RGB image.


Other notes:
 - l40: "A pioneering work in this area by Wang et al." - Not sure if pioneering is the right word given that the impact of this paper doesn't yet seem to be wide

**Questions:**

Q1) How can the method be extended to more complex datasets, less curated ones?

Q2) How crucial is video trimming?

Q3) Why do you say that the flow shown in the bottom row of Fig3 "closely resemble those of the real videos"? This seems arbitrary to me from the figure.

Q4) What is the sensitivity to the threshold epsilon? this is a hyperparameter that needs to be ablated.

---

### Official Review · Reviewer_xWHU · 2025-10-31

**Soundness:** 2
**Presentation:** 3
**Contribution:** 2
**Rating:** 4
**Confidence:** 2

**Summary:**

This paper introduces Dynamic Frame Synthesis (DFS), a novel approach for video dataset condensation. DFS progressively selects key frames with a gradient-guided synthesis mechanism and linearly interpolates the allocated frames. Results across multiple benchmark demonstrate the effectiveness of the proposed DFS method.

**Strengths:**

1. Simple yet elegant core idea: DFS adaptively include key frames based on gradient misalignment, remove the need to disentangle static and dynamic motions.
2. Comprehensive experimental results: Results across diverse benchmark validate the effectiveness of DFS. Full ablation study are conducted to provide in-depth analysis. DFS also shows potential in efficient storage.

**Weaknesses:**

1. Lack of study on hyper-parameters: The threshold $\epsilon$ is an important parameter method. It would be interesting to see how different  $\epsilon$ affects performance (across different datasets).
2. Performance gain on: From Tab.3 and 1, improvement of DFS over previous method seem very limited on UCF, Kinetics and SSv2 .
3. Following last question, previous work [1][2] show that  HMDB and UCF are biased dataset on objects and backgrounds (i.e., semantics), while SSv2 is more motion heavy dataset. I wonder if DFS truly capture the motions in videos? The update rule seems to assume smooth transition between frames, yet frames with more motion might not be included as key frame? A quantitative analysis of whether synthesized videos really capture motions would be helpful to better demonstrate the effectiveness of the method. And more ablation and analysis on SSv2 or results on other motion based dataset like Diving48 would benefit this paper.


[1] Li, Yingwei, Yi Li, and Nuno Vasconcelos. “Resound: Towards action recognition without representation bias.” *Proceedings of the European Conference on Computer Vision (ECCV)*. 2018.

 [2] Choi, Jinwoo, et al. “Why can't i dance in the mall? learning to mild scene bias in action recognition.” *Advances in Neural Information Processing Systems* 32 (2019).

**Questions:**

See weakness

---

### Official Review · Reviewer_9VCT · 2025-10-31

**Soundness:** 2
**Presentation:** 2
**Contribution:** 2
**Rating:** 4
**Confidence:** 4

**Summary:**

The paper proposes Dynamic Frame Synthesis (DFS) for video dataset condensation. DFS starts from two key frames (start/end) and linearly interpolates the rest; during optimization, it inserts new key frames only when the gradient at an interpolated frame is directionally misaligned with the gradients of its two adjacent key frames, operationalized via cosine similarity; if both cosine similarities fall below a threshold $\epsilon$, the frame is promoted to a key frame. In all experiments, $\epsilon$ is set to 0 (i.e., insert when both cosines are negative). The method uses a simple distribution-matching objective (squared L2 between mini-batch mean features) and includes warm-up/cool-down phases to stabilize insertion. Experiments on multiple datasets (miniUCF, HMDB51, Kinetics-400, and Something-Something V2) report accuracy and storage comparisons vs. coreset and prior condensation baselines.

**Strengths:**

- **Intuitive Core Insight:**
    - The gradient-based criterion for identifying where linear interpolation fails is conceptually appealing - allocating representational capacity only where motion complexity demands it.
- **Adaptive storage:**
    - Starting with 2 frames and inserting only when needed yields substantially lower storage than fixed-length approaches at similar accuracy. Storage grows sub-linearly with VPC
- **Comprehensive Ablations:**
    - The ablation studies systematically examine initialization strategies, similarity metrics, training phase components, and frame selection strategies. Each ablation shows meaningful performance deltas that support the corresponding design choices.

**Weaknesses:**

- **Inconsistent Empirical Results:**
    - On Kinetics-400 VPC 5, DFS (8.1±0.1) substantially underperforms DM (9.1±0.9), directly contradicting claims of state-of-the-art performance
    - Performance gains over baselines are often marginal with overlapping confidence intervals (miniUCF VPC 1: 17.9±0.3 vs Wang et al. 17.5±0.1; SSv2 VPC 1: 3.9±0.2 vs Wang et al. 4.0±0.1)
    - Underperformance on large-scale datasets raises serious scalability concerns that are neither explained nor addressed
- **Critical Hyper-parameter Lacks Any Justification:**
    - The threshold $\epsilon$ = 0 is the single most important hyperparameter, it determines exactly when frames are inserted. No sensitivity analysis over ε ∈ {-0.3, -0.2, -0.1, 0, 0.1, 0.2} provided. There should be an ablation about the value of $$\epsilon$$
- **Computational Cost Completely Unreported:**
    - The method requires computing gradients for interpolated candidate frames between every pair of key frames at each training step. "Low storage" does not mean cheap to condense. So, there should be analysis about runtime/computational cost etc.
- **Theoretical foundations:**
    - Lemma 1 proves only that endpoint updates cannot decrease loss at s_t (a necessary condition), but does not prove that promoting s_t to a key frame is optimal or even beneficial (no sufficiency).
    - Linear interpolation assumption (Eq. 3) for motion is physically unrealistic (ignores acceleration, easing, complex trajectories) with no empirical validation of when this holds
- **Experimental setup:**
    - Only action recognition evaluated; generalization to other video tasks unaddressed
    - Only short videos (T=8 or T=16); no evaluation on longer sequences. Is it possible to apply it longer sequences? How would incorporating longer sequences increase the cost for condensation?

**Questions:**

- Why does DFS underperform DM on Kinetics-400 VPC 5 by ~11%? This contradicts the main thesis and requires explanation.
- How many interpolated candidates are evaluated per training step? Are there statistics on this? Does it scale with current number of key frames?
- Why warm-up/cool-down fixed at 20% each? What about 10%, 15%, 30%? Any sensitivity analysis?
- What is the computational cost? Wall-clock training time, memory requirements, FLOPs compared to baselines?
- Can the method scale to longer videos (T=32, 64, 128)? How would computational cost scale?
- Can the method scale to higher resolutions (e.g., 512×512 or 1024×1024)? All experiments use 112×112 (miniUCF/HMDB51) or 64×64 (Kinetics/SSv2). How would gradient computation cost and memory requirements scale with resolution?
- Why is optical flow analysis only qualitative? Any quantitative motion fidelity metrics (flow EPE, trajectory consistency)?
- What about stronger matching objectives? Would pairing DFS frame selection with multi-moment or MMD matching improve results further?

---

### Official Review · Reviewer_1cVB · 2025-11-08

**Soundness:** 2
**Presentation:** 2
**Contribution:** 2
**Rating:** 2
**Confidence:** 4

**Summary:**

The paper introduces Dynamic Frame Synthesis (DFS), a new method for video dataset condensation that synthesizes key frames based on motion complexity. This complexity is detected via a heuristic based on the cosine similarity of gradients between an interpolated frame and its adjacent key frames. Small scale Experiments on some benchmarks are good.

**Strengths:**

1. This paper is prepared well, and the technical details are described clearly.
2. The task of video dataset condensation holds significant value for the community, as training large-scale video models requires substantial compute and storage space.

**Weaknesses:**

1. The core idea that gradient misalignment is a reliable proxy for motion complexity is an unsubstantiated heuristic method. The method's reliance on an arbitrary and hard threshold for frame insertion suggests a lack of robustness.
2. The provided theoretical support, Lemma 1, is a proof for a highly constrained and simplified scenario that does not accurately model the optimization process, thereby providing a false sense of theoretical grounding.
3. The experimental evaluation contains instances where the proposed method is outperformed by established baselines on large-scale datasets, contradicting the paper's repeated claims of achieving state-of-the-art performance.
4. All experiments in the paper are conducted with very small model and data scales. Even on Kinetics and Something-Something, only a small number of frames are used (e.g., 8), and the resolution (e.g., 64x64) is particularly low.

**Questions:**

1. The authors should consider adopting more standard experimental settings, such as an input resolution of 224×224. They should also use well-established models for experiments, such as I3D, SlowFast, and ViViT. The experimental results should also be competitive with those reported in these papers to be convincing.

---

### Note · Authors · 2025-11-13

I have read and agree with the venue's withdrawal policy on behalf of myself and my co-authors.